An assessment of sensory sensitivity in women suffering from depression using transcutaneous electrical nerve stimulation

Witkoś Joanna jwitkos@afm.edu.pl 1
Fusińska-Korpik Agnieszka 1 3
Hartman-Petrycka Magdalena 2
Nowak Agnieszka 3
1 Medicine and Health Science, Andrzej Frycz Modrzewski Krakow University , Kraków , Poland
2 Department of Basic Biomedical Science, Faculty of Pharmaceutical Sciences in Sosnowiec, Medical University of Silesia , Katowice , Poland
3 Józef Babiński Clinical Hospital in Krakow, Psychiatric Ward , Kraków , Poland
Zohar Ada
Electronic publication date: 2022 May 9
Publication date: 2022
Volume: 10
Electronic Location ID: e13373
Received 2022 Jan 6; Accepted 2022 Apr 13
Copyright: ©2022 Witkoś et al.
Copyright year: 2022
Copyright holder: Witkoś et al.
License: This is an open access article distributed under the terms of the Creative Commons Attribution License, which permits unrestricted use, distribution, reproduction and adaptation in any medium and for any purpose provided that it is properly attributed. For attribution, the original author(s), title, publication source (PeerJ) and either DOI or URL of the article must be cited.
License URL: https://creativecommons.org/licenses/by/4.0/

Keywords: Sensory sensitivity, Perception, Depression, Women, Transcutaneous electrical nerve stimulation, Electrotherapy treatment

Funding: The Andrzej Frycz Modrzewski Krakow University This work was supported by the Andrzej Frycz Modrzewski Krakow University. The funders had no role in study design, data collection and analysis, decision to publish, or preparation of the manuscript.

==============================
Background

Perception is the process or result of the process arising from the mental interpretation of the phenomena occurring, therefore it depends not only on physiology, but is also psychologically and socially conditioned. The aim of this study was to assess if there is a difference in the sensory sensitivity to an electrical stimulus in women suffering from depression and what the hedonic rating is of the lived experience of transcutaneous electrical nerve stimulation.

Methods

The depression group: 44 women, who were inpatients treated for depression at the Psychiatric Ward in the Clinical Hospital, and the control group: 41 women, matched by the age, height and weight, with no mental illness. Measures: threshold for sensing current, type of sensation evoked, hedonic rating.

Results

Median sensing threshold of electric current (depression vs. control: 7.75 mA vs. 8.35 mA; no significant), type of sensation evoked (depression vs. control: tingling 90.9% vs. 92.7%, no significant), hedonic rating (depression vs. control: unpleasant 11.4% vs. 2.4%; p = 0.003), hedonic rating (mildly ill vs. moderately ill vs. markedly ill: unpleasant 5.3% vs. 6.3% vs. 33.3%; p = 0.066).

Conclusions

Women suffering from depression exhibit a similar threshold of sensitivity to an electrical stimulus as mentally healthy women, however the hedonic rating of the stimulus acting on the skin in the group of clinically depressed women was more negative than in the mentally healthy subjects. The stimulus was described as ‘unpleasant’ for many of the mentally unhealthy women. The most negative sensations related to the electrical stimulus were experienced by women with the highest severity of mental illness according to The Clinical Global Impression - Severity Scale.

Introduction

The sensory perception process involves subjective evaluation of a stimulus acting on the human body; therefore, it is considered as an individual variable. The sensations experienced during a person’s life, including, in particular, cutaneous perception, are associated with various factors (e.g., sex, hormonal differences, genetic, social and cultural conditions) that can influence their interpretation, modification, and degree of perception (Racine et al., 2012a; Racine et al., 2012b). However, stimulus perception depends not only on physiology, but is also psychologically and socially conditioned (Landa et al., 2020). Perception is the process or result of the process arising from the mental interpretation of the phenomena occurring. These can be modified e.g., by previous experiences and individual psychosomatic conditions, including cognitive factors which receive information from the environment, store and transform it and introduce it back into the environment in the form of reactions and behaviours (Racine et al., 2012a; Racine et al., 2012b). Therefore, differences in tactile or pain sensitivity assessed only in terms of physiology are insufficient and should be combined with an assessment of the overall quality of life of the person examined. Stress, mood, and psychosocial factors contribute significantly to differences in pain perception in people (Landa et al., 2020). The available literature indicates that people “satisfied” with their lives and who have a family and social support react to external stimuli differently from those who experience so-called “interpersonal pain” and cope with a sense of rejection at different levels of functioning (Landa et al., 2020).

Physical medicine is an important component of a comprehensive physiotherapy treatment. The vast majority of physical medicine procedures are electrotherapy treatments, including transcutaneous electrical nerve stimulation (TENS) that change the sensitivity to sensory stimuli, especially pain stimuli, with the use of an electric current. The main goal of therapeutic management is pain relief, and an essential role in the process of physiotherapeutic treatment is played by neuromuscular excitability (Jauregui, Cherian & Gwam, 2016; Gladwell et al., 2016). Electrical stimuli applied during treatments may be perceived by patients as pleasant or unpleasant, even if they refer to the same type of current, e.g., TENS (Martimbianco et al., 2019; Gibson, Wand & O’Conell, 2017; Johnson et al., 2019). Therefore, it is necessary to consider what makes one person perceive a certain electrical stimulus as pleasant, while another perceives a stimulus with identical parameters using the same type of electrical current as unpleasant. In this situation, there must be additional factors that modify the perception of the stimulus by a particular person, and depression could be one of those factors.

Depression is one of the most common psychiatric disorders, associated with the highest global burden of disease (Ferrari et al., 2014), and a leading cause of disease-related disability throughout the world (GDB 2019 Diseases and Injuries Collaborators, 2020).

More than 300 million people are living with depression, with an 18% increase from 2005 to 2015 (World Health Organization, 0000). Epidemiological studies have shown that the lifetime prevalence of a major depressive disorder in women (21.3%) is almost twice that of men (12.7%). This ratio has been documented in different countries and ethnic groups (Noble, 2005).

Depression is associated with a variety of symptoms, high rates of relapse, and many physical and psychological disorders. Symptoms of depression can be psychological, physical and/or social (Sassarini, 2016; De Girolamo et al., 2020; Brockbank et al., 2021). It is characterised by a low mood, anhedonia, feelings of worthlessness and hopelessness, sadness, loss of interest or pleasure, feelings of guilt or low self-worth, feelings of tiredness, poor concentration and disturbances of sleep, appetite, and libido. Moreover, symptoms may include lack of motivation or interest, difficulty in making decisions and thoughts of self-harm or suicide, avoidance of social activities and friends and difficulty with home and family life (De Giorgi et al., 2021; Pinheiro Campos et al., 2020).

This study aimed to assess if there is a difference in the sensory sensitivity to an electrical stimulus in women suffering from depression and what the hedonic rating is of the lived experience of TENS. Therefore, the research question was: Do women who suffer from depression experience an electrical stimulus applied during electrotherapy treatments in the same way as women who are mentally healthy?

The detailed research hypotheses were as follows:

Women with depression have a different sensory threshold to TENS electrical stimuli than mentally healthy women.

Women with depression differ in terms of sensations they experienced when the TENS electrical stimulus was applied, when compared to mentally healthy women.

Women with depression differ in their hedonic rating of TENS when compared to mentally healthy women.

In women with depression, the sensory threshold, the type of sensation and the hedonic rating of TENS depend on the disease severity.

Materials & Methods

Participants

The study involved 85 women aged 18 to 49 years, of which 44 women were inpatients treated for depression at the Psychiatric Ward in the Clinical Hospital and formed the depression group (D), and 41 women were recruited from the city of Krakow and formed the control group (C). The control group consisted of the authors’ acquaintances together with members of their families who were matched by age, height and weight to the group of women suffering from depression. None of the women in the control group had any diagnosed mental illness. Detailed information on the subjects is presented in Table 1. The number of examined women was limited by Coronavirus pandemic.

Table 1 Characteristics of the women in the depression group (D) (N = 44) and control group (C) (N = 41).

	Group	Mean	St. Dev.	Me	Q1	Q2	Min	Max	U, p	
Age [years]	C	34.44	9.09	36.00	24.00	41.00	21.00	49.00	785.5, 0.308	
D	32.36	9.93	33.50	22.00	41.50	18.00	49.00	
Height [cm]	C	166.59	6.93	167.00	160.00	172.00	152.00	179.00	894.5, 0.951	
D	166.50	6.23	166.50	163.00	171.00	154.00	179.00	
Weight [kg]	C	66.46	10.13	65.00	60.00	72.00	49.00	98.00	756.0, 0.200	
D	68.41	7.99	67.50	61.50	75.00	55.00	85.00	
BMI [kg/m2]	C	24.03	3.90	24.22	21.14	25.95	18.59	34.31	766.0, 0.413	
D	24.72	3.01	24.45	22.09	27.25	18.93	32.05	
Notes.

St. Dev standard deviation

Me median

Q1 and Q3 first and third quartile

Min minimum

Max maximum

p level of statistical significance, the Mann–Whitney U test

The criteria for inclusion in the study was depression confirmed by medical diagnosis. The women were qualified for the research by the Head of the Psychiatric Ward in consultation with the attending psychologist. The exclusion criteria were the patient’s lack of consent to perform the sensory threshold test, the presence of psychotic symptoms, organic mental disorders, incapacitation (partial or complete), neurological diseases, including neuropathies and diabetes. Contraindication to the examination was also any skin diseases or damage in the place where the sensory threshold was tested.

Examination of the sensory threshold

As a sensory stimulus, an electrical impulse of increasing amplitude (intensity) was used, until the sensory threshold (minimally perceptible tingling) was reached. The current was increased slowly and gradually at 0.1 mA intervals, until the studied person informed the physiotherapist that she felt the minimum current under electrodes, then the current was not increased anymore and its value was read. The point at which this superficial sensation was felt, i.e., the sensory threshold (stimulation of thick Aβ sensory fibres), was defined as the minimally perceptible level of electrical stimulation that the subject was able to consciously feel. For this purpose, an electrotherapy device (BTL5818SLM) was used, as it allows the current intensity to be read with an accuracy of 0.1 mA. TENS with 100 Hz and 100 µs parameters and biphasic current waveform was used. The room in which the tests were performed had a constant ambient temperature of 22 °C, The subject sat on a chair, with the upper limb bent at the elbow joint and placed on a table with the forearm in supination. Surface electrodes with 12 cm2 pads, which had been dipped in warm water, were used. The electrodes were placed on the interior of the subject’s forearm on the group of flexor muscles of the wrist of the dominant hand and fastened with straps. One of the electrodes was one cm from the elbow and the other was in a straight line two cm from the first one. An important element of the examination of the sensory threshold in which the electric stimulus was used, was the fact that the same portable electrotherapy device was used to test all the women i.e., depression group and control group.

Examination of the description of the sensations

The women were additionally asked to indicate what type of sensation they felt when the current was applied. The following responses were suggested: tingling, ‘pins and needles’, scratchy, burning, stinging, numbness, and tickling.

Examination of the hedonic rating

In addition, the women were asked to give a hedonic rating to the electrical stimulus acting on the skin by indicating whether the sensation induced by the current was pleasant, neutral, or unpleasant for them. The hedonic rating of the current felt by studied people was performed for the current level that was recorded as the sensory threshold.

Procedure

The electric current parameters selected for the study resulted from their specific impact on the human nervous system. The 100 Hz frequency is a prerequisite for effective stimulation of Aβ sensory fibres. Whereas, with regard to the effect of the pulse duration on skin sensation, especially in the case of high-frequency (100 Hz) TENS stimulation, the pulse width range should generally be from about 100 µs to about 200 µs (Geng, Yoshida & Jensen, 2011; Cheing & Chan, 2009). The duration of an impulse within these limits causes a delicate and pleasant tingling sensation on the skin, and at the same time allows the effective stimulation of the nerve fibres as planned in this research. Due to the use of biphasic current waveform, it did not matter exactly where the anode (positive current pole) and where the cathode (negative current pole) were located on the above-mentioned group of muscles. The biphasic waveform of the current prevented both possible errors in the arrangement of the electrodes and a distortion of the results which can differ when stimulating the nerve fibres with the opposite poles of the current.

The research protocol has been reviewed and approved by the Bioethical Committee: permission number KBKA/65/O/2019. In accordance with the Declaration of Helsinki, before starting the study, the participants obtained information about the purpose of the study and the principles of conducting the research. They also gave their written consent to participate in the study, and were informed and made aware that they could refuse to participate in the study at any time.

Statistical analysis

The statistical analysis was performed in the Statistica 13 program. The Shapiro–Wilk test was used to statistically assess the normality of the data distribution. U Mann–Whitney and χ2 tests were used to compare the groups. The results were considered statistically significant at p < 0.05.

Results

On the basis of The Clinical Global Impression - Severity Scale (CGI-S) (Buster & Targum, 2007), the condition of 19 women suffer from depression (43.18%) was mildly ill, 16 women (36.36%) were moderately ill and nine women (20.45%) were markedly ill. The women had been treated for depression for between 1 month and 24 years, mean 6.7 years ± 6.6 years. The women were being treated with selective serotonin reuptake inhibitors (SSRIs), benzodiazepines, and lamotrigine/pregabalin.

The median value (Me) and the first and third quartiles (Q1 and Q3) of the threshold of current sensation in the group of people with depression were: Me: 7.75 mA, Q1: 6.15 mA and Q3: 9.95 mA respectively and in the control group: Me: 8.35 mA, Q1: 7.00 mA and Q3: 10.65 mA respectively (Fig. 1). The differences in the threshold of current sensation between the groups did not reach statistical significance (Mann–Whitney U test, U = 808.5, p = 0.413).

Figure 1 The threshold of current sensation in the group of women with depression (N = 44), and the control group (N = 41).

In both the Group D and the Group C, the most frequently reported sensation caused by TENS stimulation was “tingling” (Fig. 2). The percentage of women experiencing tingling after reaching the sensory threshold in the Group D was 90.91%, and in the Group C it was 92.68%. The second most frequent feeling after the activation of the electrical current was “heat/burning”, in Group D this was the response of 25.00% of the women, and in Group C 17.07% of the women. “Pinching ” was reported by around a dozen percent of women from both groups, and “numbness” was reported by only a few percent. The differences in the description of the perceived stimulus caused by the current did not differ significantly in terms of which group the woman belonged to.

Figure 2 Subjective description of the sensations caused by the electrical current in the group of women with depression (N = 44), and in the control group (N = 41).

The hedonic rating of the current-induced stimulus differed significantly between the groups (χ2 = 11.26496, df = 1, p = 0.003) (Fig. 3). For 11.36% of clinically depressed women, the perceived electrical stimulus was unpleasant, while in the Group C it was unpleasant for only 2.44%. Additionally, 19.51% of the mentally healthy women stated that the feeling caused by the current was pleasant for them.

Figure 3 Hedonic rating of the current-induced stimulus in the group of women suffering from depression (N = 44), and in the control group (N = 41) ** p = 0003.

It was also observed that in the group of clinically depressed women, depending on the degree of depression as per the CGI-S scale, there was a tendency towards a more negative hedonic rating caused by the current (χ2 = 5.43, df = 2, p = 0.066) (Fig. 4). As many as 33.33% of markedly ill women perceived the applied stimulus negatively, while in the group of mildly ill women the electric stimulus was assessed negatively by 5.26% of the women. The study also noted that the duration of the depression did not significantly affect the hedonic rating of the current-induced stimulus.

Figure 4 Hedonic rating of the current-induced sensation in mildly ill (N = 19), moderately ill (N = 16), and markedly ill (N = 9) depressed women according to the CGI-S; ## p = 0066.

Discussion

This study has shown that for women suffering from depression, this condition does not affect either the threshold for sensing the electrical stimulus, or the way the stimulus acting on the skin is described (tingling, burning, stinging). However, the interpretation of the perceived stimulus in the form of a hedonic rating is what distinguishes mentally healthy women from women suffering with depression. For women with clinical depression, the electrical stimulus applied, using TENS stimulation, was never pleasant, and the percentage of women who said the stimulus was unpleasant was higher in these women than in the mentally healthy women. It can also be observed that the percentage of women negatively perceiving electrical stimulation increased with the severity of their condition expressed in the CGI scale.

Anhedonia is one of the key features of major depressive disorder. It is mostly being considered as a decreased ability to experience pleasant feelings and influences many processes like, for example, motivation and cognition. Apart from an overall reduction in emotional response to pleasant activities and a loss of interest, it was found that anhedonia may lead to changes in perception of sensory stimuli, e.g., touch or taste (Lambert et al., 2018). When anhedonia is considered as an multifaceted construct referring to all major emotional, motivational, cognitive, sensory and social processes, it may actually explain negative assessment of the stimulus in depressive women observed in our study. As there is little research evidence for sensory impairment being one of anhedonia symptoms, further studies in the field are required (Rømer Thomsen, Whybrow & Kringelbach, 2015).

The results of our research presented above are related to a so far unexplored, yet important and worthwhile scientific issue, which cannot be overlooked when discussing this research, and which may have significant impact on the course of all physical treatment in health centres. Namely, if physical treatments are unpleasant for the patient, does the person still perceive them as therapeutic? Is physical medicine treatment effective to the same degree when the stimuli applied are unpleasant to the patient? All electrotherapy is based on ascending and descending analgesic mechanisms, but are these mechanisms activated when the procedure brings discomfort to the patient? It is known that painful stimulation of tissues and repeated pain induces hyperalgesia (Schug, Daly & Stannard, 2011), i.e., excessive sensitivity to the stimuli used. However, it is not known whether this also applies to stimuli described by patients as ‘unpleasant’. In addition, procedures in the field of physical medicine and electrotherapy are characterised by repeatability in a treatment series. Therefore, the question arises of whether the daily repetition of stimulation, that is unpleasant for a given patient, will continue to result in the analgesic effect of the electric current used during the procedures? The mechanism that leads to the perception of a stimulus by the brain is complex and is not yet fully understood. The complexity of this process results from the fact that the nervous system is not a ‘ductile’ system, but shows plasticity, which enables it to change its function under various conditions (Schug, Daly & Stannard, 2011). No answers to the questions posed above have been found yet, but it is known that in some patients physical procedures do not result in the expected analgesic effect.

Depression can be characterised by psychological and physical symptoms, is considered a public health problem and is also one of the major contributors to global disease burden (Whiteford et al., 2013). In many people, depression may accompany various painful conditions treated in primary care but go undiagnosed by a psychiatrist, because the patient does not seek this type of help. Research (Bair et al., 2003) has shown that at least 75% of primary care patients with depression present with physical complaints exclusively and seldom attribute their pain symptoms to depression or other psychiatric illness.

Depression and pain can trigger and perpetuate each other due to overlapping neuronal and emotional changes (Bair et al., 2003; Williams et al., 2003). Scientific studies show that, on average, 65% of depressed patients experience one or more painful conditions. It has also been estimated that 85% of people affected by chronic pain suffer from severe depression (Bair et al., 2003; Williams et al., 2003). The above shows that depression and pain are often co-morbid and decrease the quality of life of many individuals worldwide. An exact understanding of the pathophysiology of chronic pain and depression and of why comorbidity of these two disorders is so common is still a matter for research (Thompson et al., 2016; Nitzan et al., 2019; Malejko et al., 2021). However, although the biological mechanisms are still to be explained, it has been hypothesised that depression and pain may share the same descending tracts of the central nervous system, possibly involving serotonin, noradrenaline, and dopamine (Pinheiro Campos et al., 2020; Thompson et al., 2016). This association, called depression-pain syndrome (Cox et al., 2017), supports the concept that both conditions co-exist and exacerbate one another (Currie & Wang, 2005; Cox et al., 2017).

Depression and pain share biological pathways and neurotransmitters, so there are implications for treating both simultaneously. Depression has been found to be less effectively treated if patients experience co-morbid pain, and patients with pain are less responsive to treatment if they are suffering from depression at the same time (Pinheiro Campos et al., 2020). The presence of pain negatively affects the diagnosis and treatment of depression (Currie & Wang, 2005; Cox et al., 2017). In addition, as pain worsens, depression worsens and because pain may limit the person’s capacity to function, i.e., their ability to do their job as well as limiting any social contact, this leads to a drastic reduction in the quality of life of a person already affected by illness (Currie & Wang, 2005; Cox et al., 2017). Based on this information, it can be assumed that a significant proportion of patients who come to health centres for pain relief and to benefit from physical treatments, including electrotherapy, suffer from depression.

Moreover, it can also be assumed that in some of these people the pain is a consequence of a depressive state, and not of any actual musculoskeletal disorder.

Sensory processing refers to the organism’s ability to register and modulate sensory information (Serafini et al., 2017; Abraira & Ginty, 2013; Ulashchik, 2017), such as visual, auditory, motor, or tactile information, and to process it in order to recognise a specific stimulus (Serafini et al., 2017; Abraira & Ginty, 2013; Ulashchik, 2017). Research (Serafini et al., 2017), has found that sensory processing disorders, in particular sensory intolerance, combined with mood disorders and difficulties in adapting to specific situations, leads to maladjusted behaviour and impaired functioning in everyday life. It has been shown that sensory disorders occur in patients with affective and anxiety disorders and these are mainly expressed in the form of increased sensitivity to external stimuli, this, in turn, leads to these people avoiding any sensory impressions. In this study, it was not shown that people with depression were more sensitive to electrical stimuli compared to the group of mentally healthy women, but it was shown that TENS stimulation was unpleasant for a certain percentage of women with depression.

There is little research available in scientific reports on the perception of electrical stimuli in people with depression. Most of the published studies focus on how pain is experienced by these patients. In this research, the pain threshold was not examined as it was considered unethical. The sensory threshold test showed no difference. Examination of the hedonic rating, however, showed a tendency to a more frequent unpleasant perception of TENS stimulation in the group of women suffering from depression, especially in those with a higher degree of severity according to the CGI-S scale. This appears to be in line with other research (Serafini et al., 2017) in which it has been shown that the relationship between mood disorders and sensory processing problems is related to the severity of symptoms and the duration of the episode.

The results of existing studies on pain sensitivity in patients with depression are inconclusive. Some studies show a lower pain threshold compared to mentally healthy subjects (Lautenbacher et al., 1994; Marazziti, Castrogiovanni & Rossi, 1998), others show the opposite results (Adler & Gattaz, 1993). In research (Parker et al., 2017) it was also interesting to note that in patients with bipolar disorder, which is known to consist of depressive and manic phases, patients in the depressive phase reported reduced sensory sensitivity. This fact was not confirmed by this research, as the sensory sensitivity of mentally healthy and mentally ill women did not differ significantly. There is no doubt that the relationship between mood disorders and sensory sensitivity requires further investigation (Engel-Yeger et al., 2016).

With regard to the drugs used by the women suffering from depression in this research, it is worth noting that literature on the subject states that psychotropic drugs may affect the response to pain, i.e., antinociceptive effects (Klauenberg et al., 2008; Blier & Abbott, 2001; Bouwense et al., 2012). It is important to also take into account that the obtained study results may, to some extent, reflect the effect of medications taken by mentally ill women. Since mood stabilisers have analgesic properties, an effect on the response to sensory stimuli can also be expected. The antidepressant effect of psychotropic drugs leads to an increase in the transmission of neurotransmitters (norepinephrine, serotonin and dopamine) in cortical and subcortical structures responsible for mood regulation through, among others, a mechanism blocking the reuptake pumps of these neurotransmitters (Andrade & Rao, 2010). Among the drugs used by the women suffering from depression was, for example, pregabalin, which like some SSRIs, is sometimes used to relieve pain of central and peripheral origin. It is not clear if it has an effect on sensory sensitivity, but in research (Engel-Yeger et al., 2016) there were no differences in pain responsiveness between those patients taking and not taking SSRIs. Benzodiazepines do not change pain sensitivity, which may give a possible picture of their effect on sensory sensitivity (Klauenberg et al., 2008; Blier & Abbott, 2001; Bouwense et al., 2012).

As is known, however, pain-reactivity is not the same phenomenon as sensory-reactivity, as they are related to the stimulation of different nerve fibres. In order to be sure which drugs had or had no effect on the changes in sensory sensitivity in the women with depression in this study, an additional control group of women with depression who were not taking any drugs would have been required. This approach is impossible in a psychiatric hospital, however such study could be conducted in a sample of patients receiving various forms of psychological support in outpatient settings.

It is worth to mention here that a diagnosis of major depression in patients with pain-related symptoms is difficult, although those two conditions often co-occur (Choi et al., 2014). A patient with a mental illness is often treated in primary care only for the pain that he/she has reported to their general practitioner. In primary care facilities, the medical team, by design, focuses more on the physical ailments than on the patient’s depressed mood and mental despondency, and this is clearly the role of these specialists. However, since the results of studies by other authors (Holmes, Christelis & Arnold, 2013; Bair et al., 2003; Williams et al., 2003) show such a strong relationship between pain and depression, is it not worth paying attention to the patient more holistically? If the patient’s somatic pain is the result of psychological issues, physical treatments could be less effective. Moreover, recognizing and addressing depressive symptoms by applying a complex and interdisciplinary approach would lead to a better treatment response. Physiotherapists can offer a holistic, biopsychosocial assessment which includes an enquiry about the presence of any mental health issues, and the impact of pain on mood and morale.

The main implications of this paper, based on the Author’s findings, indicate that that TENS procedures applied in patients with depression may possibly have only a limited analgesic effect. The results of this research show a different hedonic rating in the perception of electrical stimuli in women with depression, when compared to mentally healthy women. Therefore, the fundamental question which arises is: should electrotherapeutic procedures be performed in people suffering from depression, since procedures using TENS, one of the mildest existing electrical currents used in treatment, are known to be unpleasant? As people suffering from depression informed that the stimulus was unpleasant to them at the lowest current doses that represented only the sensory threshold, it is likely that with the increase in the current to the dose representing a therapeutic stimulus it would become increasingly unpleasant or even impossible to tolerate, and result in the need to stop the procedure.

In the physical medicine, too weak stimuli acting on tissues do not have the required therapeutic effect like, for example, pain relief (Geng, Yoshida & Jensen, 2011; Cheing & Chan, 2009). The TENS stimulation intensity must be high enough to stimulate sensory fibers at a level resulting in presynaptic inhibition of pain conducted through pain fibers, i.e., closing of the control gate (Moayedi & Davis, 2013). Thus, the authors are of the opinion that TENS treatments applied at healthcare centers in people with depression may not result in the expected analgesic effect due to a limited possibility to increase the current in those patients. Therefore, it seems justified to expand the research to include those assumptions and verify whether physiotherapy procedures using TENS performed at healthcare centers in people with depression have the expected analgesic effect.

The main limitation of this study emerge from the characteristics of the group of people suffering with depression who participated in this study. Due to these limitations, the presented conclusions cannot be applied to other groups of patients suffering from depression, e.g., men and women with stage VI and VII depression on the CGI-S scale. When planning successive studies, it would be worthwhile to introduce a detailed evaluation of the ahedonia degree in studied people with depression, to determine whether it is this phenomenon that is responsible for the negative perception of the TENS stimulus in the women with depression, described in this study. It would be interesting to include typical sensory aspects of ahedonia in that evaluation. An attempt should be made to divide the study subjects into groups with primary depression and with secondary depression caused, for example, by chronic physical pain.

Moreover, it would be beneficial to conduct studies involving people who are suffering from depression of varying degrees and experiencing pain. For these people, it would be worth assessing the sensations that emerged in a complete series of TENS treatment, observing the course of the full therapy, e.g., the level of intensity achieved on individual treatment days, asking patients for a hedonic rating, and finally assessing whether the therapy had brought a satisfactory final result for the patient, as in the elimination of pain.

Additionally, in our study all women were during treatment. It would be advantageous to conduct studies in subjects with depression in whom no pharmacotherapy was initiated, and compare sensations caused by the current in those two groups of patients. It would be interesting to evaluate occurrence of possible psychosomatic symptoms in the study group and their influence on reception of the sensory stimuli and treatment effectiveness. Extending research in this area is justified.

Conclusions

In conclusion, it should be stated that women suffering from depression present a similar threshold of sensitivity as mentally healthy women to an electrical stimulus. Moreover, the sensations caused by the electric current were described by depressed women in a similar way to those given by the mentally healthy women; most often it was tingling. However, the hedonic rating of the electrical stimulus acting on the skin in the group of women suffering depression was more negative than in the mentally healthy group. The electrical stimulus was described by many of the mentally ill women as ‘unpleasant’. The electric stimulus was most negatively perceived by women with the most severe depression, i.e., those with the highest severity of illness on the CGI scale.

Supplemental Information

Supplemental Information 1 Dataset

Click here for additional data file.

Additional Information and Declarations

Competing Interests

Author Contributions

Ethics

Data Availability

The authors declare there are no competing interests.

Joanna Witkoś conceived and designed the experiments, prepared figures and/or tables, authored or reviewed drafts of the paper, and approved the final draft.

Agnieszka Fusińska-Korpik and Agnieszka Nowak performed the experiments, prepared figures and/or tables, and approved the final draft.

Magdalena Hartman-Petrycka analyzed the data, prepared figures and/or tables, and approved the final draft.

The following information was supplied relating to ethical approvals (i.e., approving body and any reference numbers):

The research protocol has been reviewed and approved by the Bioethical Committee: permission number KBKA/65/O/2019

The following information was supplied regarding data availability:

The raw measurements are available in the Supplementary Files.

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
