# Peer review of "An assessment of sensory sensitivity in women suffering from depression using transcutaneous electrical nerve stimulation"

_PeerJ, doi:10.7717/peerj.13373_

## Round 0.1 · original submission · Major Revisions

· Academic Editor

Major Revisions

Please address all the comments and concerns of the two reviewers, and then in the rebuttal letter list each in turn.

Reviewer 1 ·

Basic reporting

This is, in summary, an interesting paper aimed to investigate whether women who suffer from depression experience an electrical stimulus applied during electrotherapy treatments in the same way as women who are mentally healthy. The authors found that Median sensing threshold of electric current (depression vs. control: 7.75mA vs. 8.35mA; no significant), type of sensation evoked (depression vs. control: tingling 90.9% vs. 92.7%, no significant), hedonic rating (depression vs. control: unpleasant 11.4% vs. 2.4%; p=0.003), hedonic rating (mildly ill vs. moderately ill vs. markedly ill: unpleasant 5.3% vs. 6.3% vs. 33.3%; p=0.066).

Experimental design

As the most relevant aims/objectives of this study have been reported, the main hypotheses underlying the present study could be briefly specified by the authors.

In addition, the conditions of the 19 women suffer from depression need to be described within the Results section rather than within the Methods section as in the current version of the manuscript.

Validity of the findings

The authors should present and discuss, in the first lines of the Discussion section, the most relevant study findings of this paper instead of focusing on the main literature gaps upon the main topic that have been adequately stressed within the main text.

Importantly, the most relevant study limitations/shortcomings should be described in a more detailed manner as the the main caveats have been only partially reported.

Additional comments

Finally, what is the take-home message of this paper? While the authors reported that females suffering from depression present a similar threshold of sensitivity as mentally healthy females to an electrical stimulus, they failed, in my opinion, to focus on some conclusive remarks to this specific regard. Specifically, what are the main implications of this paper according to these findings? How the mentioned results may be generalized? Here, some additional details might be useful for the general readership and should be provided by the authors based on their expertise.

Reviewer 2 ·

Basic reporting

Some improvement in the writing and referencing is required , but this is essentially a well-written paper which needs some revision to overcome some problems, which are detailed below.

Experimental design

The design is good but some improvement in the description of the methods is required.

Validity of the findings

The findings are valid, interesting and increase our understanding about TENS but some of the statements in the discussion include unreferenced over-generalisation and should be improved.

Additional comments

I think that this is an interesting study with new insights into the experience of potential TENS users. The findings could help to explain limited benefits from TENS in patients with depression. However some of the detail requires improvement, and I have provided a detailed description of these problems below.
Line 21: “Beckground.” This is a typo, it should be Background.

Lines 21-23: “Perception is the result of experiences arising from the mental
interpretation of the phenomena occurring, therefore it depends not only on physiology or biology, but is also psychically conditioned.”
Please could you review the meanings of the words experience, perception and psychically? It could be possible to state that our lived experiences are the result of perception (if perception is the process by which sensations are processed, organised and interpreted). Perception also has a dual meaning, describing both the process and the end result of the process, and this second meaning is equivalent to “experience”.
These definitions might be helpful: https://dictionary.apa.org/perception and https://dictionary.apa.org/experience

I suspect that you mean “psychologically conditioned” not psychically conditioned, but do check: https://dictionary.cambridge.org/dictionary/english/psychically

Physiology is a subset of biology, so you could perhaps remove one of these words.

Line 44: “Sensory perception is based on the subjective evaluation of a stimulus acting on the
human body, thus it is an individual variable.”
Please could you review this sentence, based on the comments made above about the definition of perception as a process? Is part of the process of sensory perception separate from subjective experience, for example could we identify processing of TENS stimulation within the nervous system of a person who was asleep, and therefore this aspect is not subjective? Is it possible that the subjective evaluation is the final stage of sensory perception? Some psychologists (e.g. Albert Bandura) conceptualise subjective experience as an emergent property of biological systems, and this concept might be helpful here. A reference is: https://www.annualreviews.org/doi/abs/10.1146/annurev.psych.52.1.1

Line 45: “psychically” here is probably best changed to “psychologically”.

Lines 55-57: “Therefore, differences in tactile or pain sensitivity assessed only in terms of physiology are insufficient and must be obligatorily combined with an assessment of the overall "satisfaction with life" of the person examined.”
The word obligatorily could be removed. I’m not sure that “satisfaction with life” is the best term to use here: “quality of life” however is a term that is commonly used in terms of patient experience and health outcomes. Would this work better?

Lines 75-77: “Depression is one of the most common psychiatric illnesses, as well as being both a major cause of diseases and a leading cause of disease-related disability throughout the worldwide.”
I think the statement that depression is a major cause of diseases should be referenced.
I think the word “worldwide” should be changed to “world”.

Lines 93-96: “This study aimed to assess if there is a difference in the sensory sensitivity to an electrical stimulus in women suffering from depression and what the hedonic rating is when the stimulus acts on the organism’s tissues.”
I think this could be altered to focus more on the TENS experience, such as:
“This study aimed to assess if there is a difference in the sensory sensitivity to an electrical stimulus in women suffering from depression and what the hedonic rating is of the lived experience of TENS.”

Line 100: were the women inpatients, outpatients or both?

Line 107: I think that the Clinical Global Impression - Severity Scale should be referenced.

Line 139: I have not been able to find an explanation of the intensity of the TENS stimulation when the participants were asked to make a hedonic rating. What instructions were given regarding turning up the TENS intensity?

Line 143: “organism’s tissues” could perhaps be replaced by the words “human nervous system”.

Line 195-6: “Moreover, it seems that in the scientific field there is no direct practical
references to the effects of electrotherapy in people suffering from depression.”
I think that there are references to this: you could review papers reporting clinical audits such as Lampl et al 1998: https://journals.lww.com/clinicalpain/Abstract/1998/06000/Transcutaneous_Electrical_Nerve__Stimulation_in.8.aspx

I tried a search of Google Scholar on “transcutaneous electrical nerve stimulation AND depression” and found several papers of interest on this subject.

Line 209-10: “Namely, if physical treatments are unpleasant for the patient, does the organism still perceive them as healing?”
I suggest that this is reworded to:
Namely, if physical treatments are unpleasant for the patient, does the person still
perceive them as therapeutic? I don’t think we should be calling people “organisms” (even if we are!), and I don’t think that TENS is “healing” even if it might be therapeutic.

Line 211-13: “All electrotherapy is based on ascending and descending analgesic mechanisms, but are these mechanisms activated when the procedure brings discomfort to the patient?”
I think that mentioning secondary hyperalgesia at this point might be helpful.

Line 239: The paper by Leo is quite old now (2005) and I think a more up to date reference should be included.
I spent a minute or two looking for newer articles, and there are plenty. This review might be a good start: https://www.iasp-pain.org/publications/relief-news/article/depression-pain-changes/

Lines 255-7: “It is highly probable that people suffering from depression and who are being given physical treatments, do not divulge information about their mental health to the physiotherapist.”
I think that this is a significant assertion and is not referenced. Can you provide evidence for this statement? Physiotherapists can offer a holistic, biopsychosocial assessment which includes an enquiry about the presence of any mental health issues, and the impact of pain on mood and morale.

Line 257: “…any actual somatic disorder.”
It could be argued that patients who experience pain associated with depression are experiencing a somatic disorder, especially if we think about the mind and body being integrated rather than a mind-body dualist view. I wondered if replacing “somatic” with “musculoskeletal” might be helpful here?


Lines 275-7: “The sensory threshold test, however, showed a tendency to a more frequent unpleasant perception of TENS stimulation in the group of women suffering from depression, especially in those with a higher degree of severity according to the CGI-S scale.”
I think that this needs to be corrected, as the sensory threshold test showed no difference. I think you mean the hedonic rating?

Lines 296-8: “The effect of psychotropic drugs on the human organism is mainly based on the reuptake of norepinephrine and serotonin, increasing the levels of these neurotransmitters (Klauenberg et al., 2008).”
This seems to me to be a significant over-generalisation. Is this really true for ALL psychotropic drugs? See the statement in line 307 about benzodiazepines.

Line 303: “as well as unacceptable for health reasons.”
Not all depressed people take anti-depressants, due to lack of efficacy and intolerable side-effects. It should be possible to recruit from this population.

Line 311-3: “Unfortunately, pain has a negative impact on the diagnosis of depression, as it often moves to the fore so much that it completely captivates the attention of doctors and physiotherapists.”
This is another significant over-generalisation. Can you provide evidence to support this statement? I suggest that you search the literature on secondary depression due to pain.

Lines 317-19: “If the patient's somatic pain is the result of psychological issues, it is very unlikely that physical treatments will bring a particular therapeutic effect.”
Is it VERY unlikely? What is the evidence for this statement? We know that depression can affect treatment outcomes, but within that there is a complex picture of variable outcomes for different patients. Some will have primary depression, others secondary depression due to pain.

Line 342: “Group D” might be better stated as “depressed women” in the conclusion.

General point regarding the discussion. Anhedonia is a key feature of depression. Many of these depressed women will be experiencing reduced pleasure in many experiences which would normally be pleasant. This might apply (for example) to social contact, music, exercise and chocolate. The reduced hedonic rating of TENS might just be part of this general phenomena: I cannot find any discussion of this key point.

Figure 4: “markedely” should be corrected to “markedly”

---

## Round 0.2 · accepted · Accept

· Academic Editor

Accept

Thank you for doing such a good job on the revision.

Reviewer 2 ·

Basic reporting

The authors have responded constructively to the review and have produced a paper which is clear, well referenced and well structured.

Experimental design

The authors have clarified the sections in the methods which were picked up in the first review, and this section is now satisfactory.

Validity of the findings

The authors have again responded well to the first review, and the findings and interpretation are now more robust.

Additional comments

I am satisfied with the second draft, and would support its publication.